# Therapeutic Advances of Rare ALK Fusions in Non-Small Cell Lung Cancer

Yan Xiang 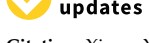, Shiyu Zhang, Xiaoxu Fang, Yingying Jiang, Tingwen Fang, Jinwen Liu and Kaihua Lu *

Department of Oncology, The First Affiliated Hospital of Nanjing Medical University, Nanjing 210029, China
* Correspondence: lukaihua@njmu.edu.cn; Tel.: +86-13605179453

**Abstract:** Non-small cell lung cancer (NSCLC) accounts for approximately 85% of all lung cancer cases and is the leading cause of cancer-related death. Despite advances in chemotherapy and immunotherapy, the prognosis for advanced patients remains poor. The discovery of oncogenic driver mutations, such as anaplastic lymphoma kinase (ALK) mutations, means that a subset of patients has opportunities for targeted therapy. With the improvement of genetic testing coverage, more and more ALK fusion subtypes and ALK partners have been discovered, and more than 90 rare ALK fusion subtypes have been found in NSCLC. However, unlike the common fusion, echinoderm microtubule-associated protein-like 4 (EML4)-ALK, some rare ALK fusions such as striatin (STRN)-ALK and huntingtin interacting protein 1 (HIP1)-ALK, etc., the large-scale clinical data related to its efficacy are still immature. The clinical application of ALK-tyrosine kinase inhibitors (ALK-TKIs) mainly depends on the positivity of the ALK gene, regardless of the molecular characteristics of the fusion partner. Recent clinical studies in the ALK-positive NSCLC population have demonstrated differences in progression-free survival (PFS) among patients based on different ALK fusion subtypes. This article will introduce the biological characteristics of ALK fusion kinase and common detection methods of ALK fusion and focus on summarizing the differential responses of several rare ALK fusions to ALK-TKIs, and propose corresponding treatment strategies, so as to better guide the application of ALK-TKIs in rare ALK fusion population.

**Keywords:** anaplastic lymphoma kinase; tyrosine kinase inhibitor; rare fusion; non-small cell lung cancer

## 1. Introduction

Anaplastic lymphoma kinase (ALK) mutation plays an important role in the occurrence and development of non-small cell lung cancer [1,2]. The ALK gene encodes a single transmembrane receptor tyrosine kinase that belongs to the insulin receptor superfamily [3–5], which has an extracellular domain, a transmembrane segment, and a cytoplasmic receptor kinase segment [6]. Studies have identified pleiotrophin, midkine, and heparin as putative ALK ligands [7,8]. Usually, the ligand binds to the extracellular domain, promotes the coupling of two adjacent ALK proteins on the cell membrane, then activates the intracellular signal pathways such as RAS-MAPK, PI3K-AKT, JAK-STAT, MEKK2/3-MEK5-ERK5, CRKL-C3G and promotes cell growth [9,10].

So far, four detection methods and six ALK tyrosine kinase inhibitors (ALK-TKIs) have been approved for clinical use. As far as ALK gene detection is concerned, fluorescence in situ hybridization (FISH) is the preferred gold standard method; immuno-histochemistry (IHC) can be used for screening because it is simple and cheap, and the FDA-approved antibody (Ventana D5F3) can be used to detect ALK fusion independently without FISH validation. Reverse transcription polymerase chain reaction (RT-PCR) can also be used, but it cannot detect some rare fusions. Next-generation sequencing (NGS) can not only detect ALK fusion but also determine the type of fusion and other accompanying driver genes, which meets the current clinical needs. The emergence of ALK-TKIs has completely

changed the treatment strategy and prognosis of advanced NSCLC patients with ALK fusion [11,12]. The first-generation targeted drug simultaneously has three targets: ALK, ROS1, and C-MET. The PROFILE 1014 study demonstrated that, compared with standard platinum-based chemotherapy, crizotinib significantly prolonged the median progression-free survival (mPFS, 10.9 months vs. 7.0 months; $p < 0.001$) and objective response rate (ORR, 74% vs. 45%; $p < 0.001$) of previously untreated patients with ALK-positive advanced NSCLC [13]. The second-generation ALK inhibitors can be used for crizotinib resistance such as L1196M and C1156Y mutations [14]. Lorlatinib, the third-generation targeted drug for ALK, can inhibit almost all resistance mutations that lead to crizotinib resistance, such as G1202R, G1202del, etc. [15].

Although ALK fusion is a clinically proven tumor therapeutic target, compared with other carcinogenic drivers, such as epidermal factor growth receptor (EGFR), how to better target ALK fusion lacks accuracy. More than 90 different ALK fusion partners have been identified in lung cancer, including EML4, STRN, KIF5B, HIP1, TPM-3/−4, DCTN1, GCC2, TFG, etc. [16–23]. Several clinical trials have demonstrated the efficacy of ALK-TKIs in EML4-ALK-positive NSCLC [24]. The Alex study reported that regardless of the EML4-ALK variant, the progression-free survival (PFS) of untreated ALK-positive NSCLC treated with alectinib was better than that of crizotinib [25]. However, unlike the classic EML4-ALK, some rare fusions such as STRN-ALK and HIP1-ALK, etc., the large-scale clinical data related to efficacy are still immature. The clinical application of ALK-TKIs mainly depends on the positivity of the ALK gene, regardless of the molecular characteristics of the fusion partner. Recent clinical studies in patients with ALK-positive NSCLC have shown that there are differences in PFS based on different subtypes [26]. Therefore, understanding the responses of different rare ALK fusions to ALK-TKIs is necessary to guide clinical treatment. This article will introduce the biological characteristics of ALK fusion kinases and the common detection methods of ALK fusion and focus on summarizing the rare fusions other than EML4-ALK and their treatment progress, and propose corresponding treatment strategies, so as to better guide the application of ALK-TKIs in rare ALK fusion population.

## 2. The Biology of ALK Fusion Kinases

It is generally believed that the formation of pathogenic fusion genes requires three steps (Figure 1A): firstly, external factors (such as various physical, chemical and biological factors) or internal mechanisms of cells cause DNA double-strand breaks; secondly, the ends of the broken DNA are close to each other; thirdly, the DNA ends are aberrantly repaired, probably by alternative non-homologous end-joining [27]. DNA junctions often show short homology, called microhomology, which allows non-homologous ends to be connected. In the last step, the expression of the fusion gene gives the cell growth and/or survival advantages, so as to achieve selective cloning and amplification [28].

ALK fusion is the result of chromosome rearrangement between the ALK gene and other genes (Figure 1B). One of the most common types of rearrangement is interchromosomal translocation, which involves the exchange of chromosomal material between heterologous chromosomes, such as TFG-ALK, KIF5B-ALK, etc. [29,30]. Intrachromosomal rearrangement is also usual, especially with paracentric inversion (excluding centromere). For example, EML4-ALK involves the short arm of chromosome 2 [31]. Two other intrachromosomal rearrangements are deletions and duplications, such as STRN-ALK (partial sequence deletion of the short arm of chromosome 2 results in fusion of STRN exon 3 and ALK exon 20) [32], C2orf44-ALK (caused by a 5.2-megabase pair tandem duplication on chromosome 2) [33]. However, the production of ALK fusion protein needs other conditions. Firstly, the ALK gene breakpoint must include the entire tyrosine kinase domain (usually at exon 20). Secondly, the promoter region tends to be derived from the fusion partner, probably due to the fact that the ALK promoter is inactive in adults and thus cannot drive gene transcription [34]. Finally, the fusion partner must contain the oligomerization domain. Most fusion partners contain coiled coils or leucine zipper domains that drive fusion kinase activation [1,35].

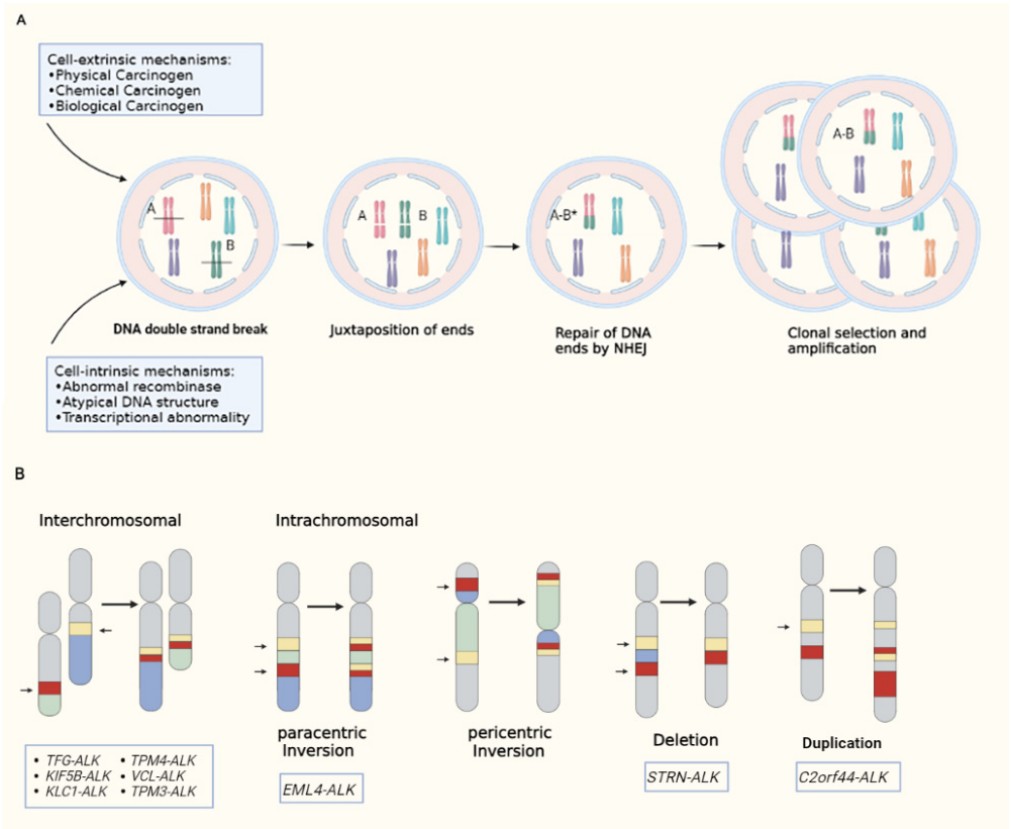

**Figure 1.** (**A**): Generation of pathogenic fusion genes; (**B**): types of ALK rearrangement.

The molecular structure and biochemical properties of fusion proteins can affect the differential response of patients to ALK-TKIs, including: (1) the type of oligomeric domains; (2) the stoichiometry of oligomerization: dimers, trimers, and multimers [36–39]; (3) the intrinsic kinase activity of the fusion protein [40]; (4) protein–protein interactions, which vary mainly by the structure of the fusion partner; (5) length of the 5′ partner; (6) protein folding and spatial structure; (7) protein stability. For example, EML4-ALK variants 1, 2, and 3 have different structures. EML4-ALK variant 3 lacks the HELP and WD40 domains, while the WD40 domain of EML4-ALK variant 2 has 5 WD40 repeats less than that of variant 1, resulting in differences in protein stability and tyrosine kinase activity among the three variants [41]. In one study, Yoshida et al. [42] reported differential responses to crizotinib in 35 patients with NSCLC harboring different EML4-ALK variants. The results showed that EML4-ALK variant 1 showed better efficacy compared with non-variant 1, with the objective response rate (ORR) of 74% versus 63% ($p = 0.7160$, no significant difference), disease control rate (DCR) 95% versus 63% ($p < 0.05$, significant difference) and the PFS11 months versus 4.2 months ($p < 0.05$, significant difference).

## 3. Detection Methods for ALK Rearrangements

### 3.1. Immuno-Histochemistry

IHC is mainly detected by the binding reaction between high-sensitivity ALK antibody and antigen, combined with signal cascade amplification (Figure 2). Normal lung tissue is difficult to express detectable levels of ALK, but the level of ALK expressed by rearranged ALK-positive NSCLC is moderate [43–45], and the combination of highly sensitive ALK antibodies makes IHC quite reliable in detecting ALK-positive NSCLC [43,46]. However, due to the heterogeneity of tumor cells and the uneven expression of target proteins, IHC detection may be false negative in some cases. In addition, IHC cannot identify the type and molecular structure of fusion partners, which may have an impact on treatment [26]. Therefore, clinicians recommend the use of IHC for preliminary screening [47].

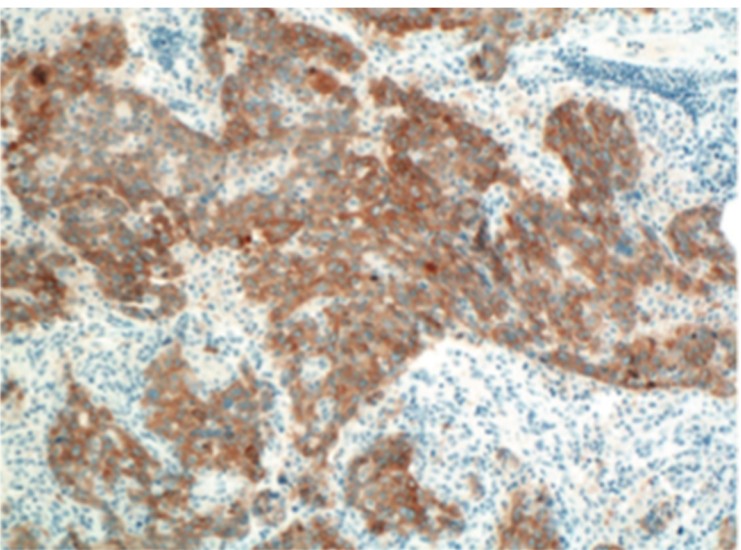

**Figure 2.** Immuno-histochemistry (IHC); From Roche Ventana.

*3.2. Fluorescence in Situ Hybridization*

FISH is to use specific nucleic acid probes labeled with fluorescence to hybridize with corresponding DNA molecules in cells, and then observe the fluorescence signal under the microscope to determine the position of DNA molecules binding to specific fluorescent probes in chromosomes [48,49] (Figure 3). FISH is a very sensitive and rapid method, which makes up for the false negative defect of IHC due to different expression intensities of ALK protein [50]. However, for some specific rearrangements, such as the separation of EML4 and ALK on chromosome 2p by only 12.5 Mb, it is impossible to accurately determine whether the two signals are separated under the microscope [51]; In addition, like IHC, FISH can only be used to determine whether the ALK site is broken, and cannot distinguish the type of fusion partner. Despite these deficiencies, FISH remains the gold standard for the detection of ALK rearrangements [52].

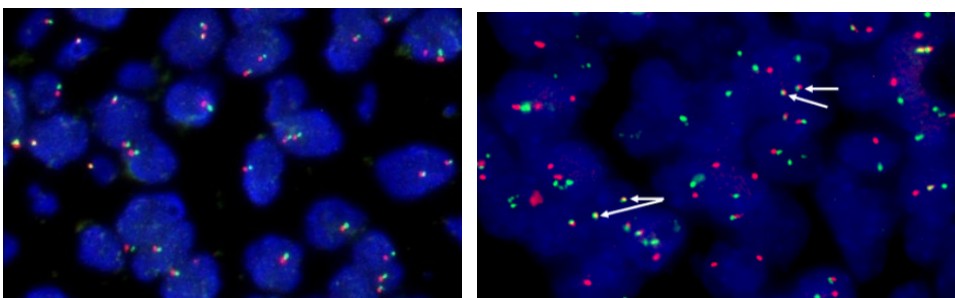

**Figure 3.** FISH interpretation criteria: (From Abbott Molecular, Vysis probes) (I) Count 50 tumor cells: If the number of positive cells was ≥50% (25), the patient was judged to have ALK rearrangements (**On the right**). If the number of positive cells was less than 10% (5), the patient was judged to have no ALK recombination (**On the left**); (II) If the positive cells were ≥10% (5) and <50% (25), another 50 tumor cells were counted, and then the 100 tumor cells were counted together. If the number of positive cells is ≥15%, the patient is judged to have ALK recombination. If the number of positive cells was less than 15%, the patient was judged to have no ALK recombination. Note: The positive cells were the cells with orange and green signal separation or orange signal alone.

*3.3. Reverse Transcription Polymerase Chain Reaction*

RT-PCR detects fusion mutations by extracting total RNA from tissues or cells, using the mRNA as a template, and then using pre designed primers to reverse transcribe sample RNA(Figure 4). Clinical studies have shown that different ALK fusion partners

may affect the dimerization of fusion kinases, resulting in differences in tumor biological characteristics [53]. Therefore, it is important to identify specific fusion partners before selecting appropriate treatment measures. RT-PCR can sensitively detect the type of ALK fusion partner, and it is also applicable to some specimens that are not suitable for slice preparation, such as bronchial lavage fluid, pleural effusion, or blood [54]. However, the accuracy of RT-PCR diagnosis largely depends on the RNA quality of samples [54,55]. Before the successful identification of ALK fusion partners, many different primer sets need to be used, and unknown fusion variants cannot be detected [56,57].

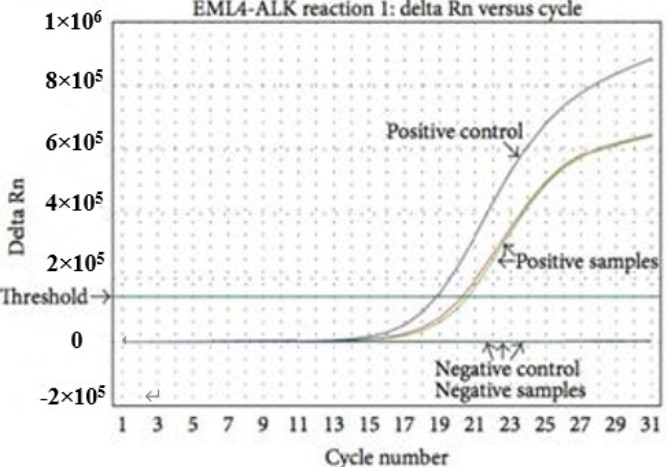

**Figure 4.** Realtime RT-PCR; From AmoyDx EML4-ALK Fusion Gene Detection Kit.

*3.4. Next-Generation Sequencing*

NGS can simultaneously detect hundreds of gene mutations, insertions, deletions, fusions, copy number variations, etc., providing an effective and accurate alternative for FISH testing to detect known and new ALK fusions. Compared with traditional pathological detection, NGS has the advantages of high efficiency, high throughput and short cycle (Figure 5). As mentioned above, not all ALK fusions have biological significance. In order to become an oncogene, the fused ALK gene needs to retain its own kinase domain and the original correct reading frame. NGS can clearly observe the breaking site, so it can clearly know whether the fusion gene formed has normal biological function. In addition, NGS can also predict the therapeutic effect and drug resistance mechanism of drugs by detecting circulating tumor DNA (ctDNA) and circulating free DNA (cfDNA) in the blood, which is expected to improve the prognosis [58].

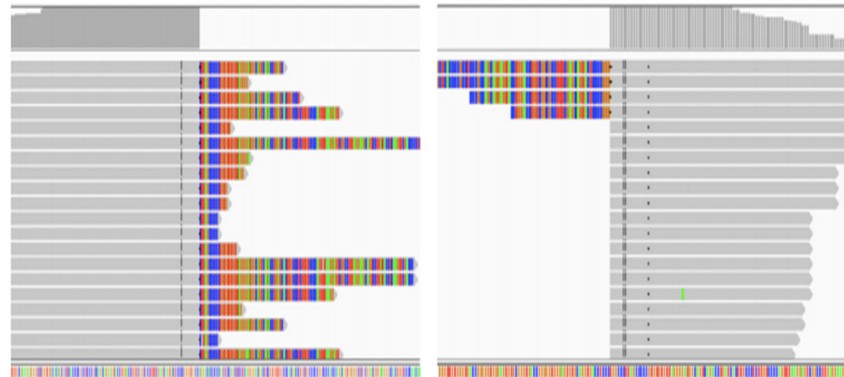

**Figure 5.** High-throughput sequencing Kit (next-generation sequencing technology, NGS).

## 4. Rare ALK Fusions and Therapeutic Advances

### 4.1. STRN-ALK

The STRN-ALK fusion in NSCLC was first described by Majewski et al. in 2013 using RNA sequencing [59]. Striatin (STRN) is a protein encoded by the STRN gene, including STRN, STRN3, and STRN4. Members of the striatin family feature multiple protein-binding domains, such as a caveolin-binding motif, a coiled-coil structure, a calmodulin-binding site, and a WD-repeat domain [60]. The STRN-ALK fusion has been reported to involve an intrachromosomal translocation of exon 3 of STRN and exon 20 of ALK within the short arm of chromosome 2 (S3A20, separated by ~7.5 Mb) [32,61,62]. STRN induces constitutive activation of ALK kinase through dimerization mediated by the 5′ coiled-coil domain in the gene [32]. Previous studies have shown that STRN-ALK can affect the aggressive characteristics of tumors, including lymph nodes, and distant metastases [32,63,64].

More than 20 cases of STRN-ALK fusion have been reported worldwide, including thyroid carcinoma, NSCLC, colorectal cancer, and renal cancer [62,65–67]. Among them, there were seven cases of NSCLC (Table 1). Four patients received first-line treatment with alectinib [18,64,68,69], three patients responded well to alectinib, while the other developed within 3 months [64]. Through the study of this progressive case, it was found that the patient was also accompanied by overexpression of ATP binding cassette subfamily B member 1 (ABCB1) mRNA, and this mechanism was proved to be a potential drug resistance factor of ALK-TKIs [70], so this patient responded poorly to alectinib. Li et al. reported a patient with stage IIIA ALK-positive lung adenocarcinoma who was treated with alectinib after operation [71]. After 3 months, multiple lung progression occurred. NGS showed a rare STRN-ALK fusion with MET amplification. The patient then received second-line treatment with crizotinib and achieved partial remission (PR) one month later. PFS exceeded 11 months. It has been proved that MET amplification is a recurrent drug resistance mechanism of the second-generation ALK-TKIs [15,72]. Given the high selectivity and strong affinity of alectinib for ALK fusions, crizotinib exhibits a relatively low affinity for ALK and can target multiple tyrosine kinases such as MET and ROS1 [73,74]. The progress of this patient is likely due to the occurrence of acquired MET amplification. In contrast, patients with STRN-ALK benefit more significantly from second-line or multi-line treatment with crizotinib. Zhou et al. reported a case of stage IV NSCLC patients with acquired resistance mutation of STRN-ALK after receiving osimertinib [75]. The researchers gave gefitinib combined with crizotinib in combination with gene testing results and PFS lasted for more than 6 months, which once again proved the feasibility of crizotinib in the treatment of STRN-ALK fusion.

**Table 1.** Published details of STRN-ALK-positive NSCLC patients treated with ALK-TKIs.

|  | Ref | Year | Accompanying Mutations | ALK-TKIs | Treatment | Response |
|---|---|---|---|---|---|---|
| 1 | Yang, Y. et al. [76] | 2017 | MYC amplification; TP53 (R181C) | Crizotinib | Third line | CR > 6 m |
| 2 | Zhou, C. et al. [75] | 2019 | EGFR (19DEL) | Crizotinib +Gefitinib | Third line | PR > 6 m |
| 3 | Nakanshi, Y. et al. [64] | 2017 | ABCB1 mRNA overexpression | Alectinib | First line | PD < 3 m |
| 4 | Su, C. et al. [18] | 2020 | GRM8 (E508K); SETD2 (E1553K) | Alectinib | First line | PR > 19 m |
| 5 | Nagasaka, M. et al. [69] | 2020 | TP53 (L43fs);MYC amplification | Alectinib | First line | PR > 6 m |
| 6 | Zeng, H. et al. [68] | 2020 | PDK1-ALK (P7: A20); TP53 | Alectinib | First line | PR > 7 m |
| 7 | Li, M. et al. [71] | 2021 | MET amplification | Alectinib Crizotinib | First line Second line | PD: 3 m PFS > 11 m |

### 4.2. KIF5B-ALK

The KIF5B gene is located on the short arm of human chromosome 10 and encodes the kinesin family 5B gene (KIF5B). The KIF5B protein is the main component of the

microtubule-associated motor protein complex, which mediates the transport of organelles in eukaryotic cells [29,77]. Exons 1 to 24 of KIF5B can fuse with exon 20 of ALK to produce a new fusion gene KIF5B-ALK, which mediates ALK dimerization through the domain of KIF5B, thereby activating its tyrosine kinase activity [29]. As one of the rare fusion partners of ALK, KIF5B only accounts for about 0.4% of ALK fusions. Studies have shown that KIF5B-ALK transfected cells have significantly enhanced proliferation, migration, and invasion [16].

Takeuchi et al. reported a patient with lung adenocarcinoma harboring KIF5B-ALK fusion, in which intron 24 of KIF5B was fused to intron 19 of ALK [29] (Table 2). Wong et al. reported another variant of the KIF5B-ALK fusion, KIF5B exon 15 fused to ALK exon 20 [16]. However, neither case reported the sensitivity of KIF5B-ALK to ALK-TKIs. Zeng et al. introduced a case of lung adenocarcinoma with rare KIF5B-ALK (intron 20 of KIF5B is connected to intron 20 of ALK) and obtained PFS for 11 months after treatment with crizotinib [78]. In addition, NGS-ctDNA genomic analysis after craniocerebral progression in the patient suggested the known KIF5B-ALK fusion and ALK exon 23 L1196M missense mutation. The patient immediately received second-line treatment with ceritinib, craniocerebral lesions were significantly reduced and a 9-month PFS was achieved with continuous follow-up. Although the initial effect of crizotinib is remarkable, patients inevitably develop resistance. Clinical studies have shown that 66.7% of patients have secondary mutations in the ALK kinase domain after treatment with crizotinib. These drug-resistant mutations include L1196M, G1269A, G1202R and C1156Y [79–82]. Among them, L1196M is the most common secondary mutation in NSCLC. It is located in the protein kinase domain of ALK protein and can control the entry of small molecule ALK inhibitors into the hydrophobic pocket within the catalytic site, thereby sterically blocking the binding of crizotinib to ALK [83]. Second-generation ALK-TKIs are known to be effective against ALK-related secondary resistance mutations. Another preclinical evaluation showed that ceritinib can overcome crizotinib-induced resistance mutations, especially the secondary L1196M mutation [84]. Therefore, patients with KIF5B-ALK rare fusion can flexibly combine the first-, second-, and third-generation ALK-TKIs for sequential therapy in combination with ALK fusion type and mutation type, so as to improve the prognosis of patients with ALK-positive NSCLC.

**Table 2.** Published details of KIF5B-ALK-positive NSCLC patients treated with ALK-TKIs.

| | Ref | Year | Variants | ALK-TKIs | Treatment | Response |
|---|---|---|---|---|---|---|
| 1 | Takeuchi, K. et al. [29] | 2009 | (K24:A19) | Not treated with ALK-TKIs | – | – |
| 2 | Wong, D.W. et al. [16] | 2011 | (K15:A20) | Not treated with ALK-TKIs | – | – |
| 3 | Zeng, H. et al. [78] | 2021 | (K20:A20) | Crizotinib Ceritinib | First line Second line | PFS: 11 m PFS > 9 m |

### 4.3. HIP1-ALK

Huntingtin interacting protein 1 (HIP1) contains multiple domains, and its N-terminal homologous domain can bind to inositol polyphosphate signal, playing an important role in clathrin-mediated receptor transport and cell survival [85,86]. Studies have shown that HIP1 can be overexpressed in various human tumor cells and promote their clonal proliferation. Kalchman et al. reported a novel HIP1-ALK fusion gene in NSCLC for the first time [87]. The HIP1-ALK protein contains the coiled-coil domain of HIP1 and the near membrane intracellular region of ALK. Through the dimerization of the coiled-coil domain, the activity of ALK tyrosine kinase is abnormally activated, resulting in the occurrence of tumor [12,88].

A total of five HIP1-ALK variants have been reported so far: (H28:A20); (H21:A20); (H19:A20); (H22:A21) and (H30:A20) [20,24,89–94] (Table 3). Clinical studies have shown that HIP1-ALK-positive patients carrying different variants have significantly different

responses to ALK-TKIs. Fang et al. tested the sensitivity to crizotinib by establishing a patient-derived xenograft (PDX) model of NSCLC with HIP1-ALK fusion (H28:A20) in vitro [93]. Unfortunately, this PDX model is derived from moderately differentiated squamous cell carcinoma, and it has not been further validated whether the patients have clinical benefits. On this basis, Couëtoux et al. successfully that lung adenocarcinoma patients with HIP1-ALK (H28:A20) responded well to crizotinib, and the PFS could reach 26.9 months [94]. In addition, Hong et al. reported a case of a postoperative patient with HIP1-ALK fusion (H21:A20) who received adjuvant crizotinib and did not experience recurrence or metastasis after 15 months of follow-up [20]. However, some studies have shown that HIP1-ALK patients with variants such as (H19:A20) and (H30:A20) have a poor response to crizotinib, in which Li, M. et al. found that HIP1-ALK (H19:A20) primary resistance to crizotinib and subsequent second-line treatment with alectinib resulted in PFS of more than 9 months [24]. Given that, Ou, S.H. et al. reported that HIP1-ALK (H30:A20) was ineffective against crizotinib [90]. Li Y et al. proved that HIP1-ALK (H30:A20) positive patients can benefit from alectinib with a PFS of more than 19 months [91]. Another retrospective study assessed clinicopathological features, genomic features, responses to ALK-TKIs, and resistance mechanisms in 11 patients with HIP1-ALK fusion from China [95]. The ORR of 10 patients treated with crizotinib was 90% [9/10 cases, 95% confidence interval (CI): 54.1%–99.5%], the mPFS was 17.9 months [95% CI: 5.8-NA], and the median overall survival (mOS) was 58.8 months (95% CI: 24.7-NA). One patient receiving first-line treatment with lorlatinib achieved partial response (PR) for more than 26.5 months, of the ten patients who received crizotinib, four underwent biopsy after progression, and two carried acquired ALK mutations (L1152V/Q1146K and L1196M). Although the HIP1-ALK fusion initially responds well to crizotinib, resistance is inevitable, whereas brigatinib is effective in patients who have failed crizotinib due to the L1152V/Q1146K resistance mutations, which may be related to the high affinity of brigatinib with these mutations.

**Table 3.** Published details of HIP1-ALK-positive NSCLC patients treated with ALK-TKIs.

| | Ref | Year | Variants | ALK-TKIs | Treatment | Response |
|---|---|---|---|---|---|---|
| 1 | Fang, D.D. et al. [93] | 2014 | (H28:A20) | Not treated with ALK-TKIs | – | PDX is sensitive to crizotinib |
| 2 | Hong, M., et al. [20] | 2014 | (H21:A20) | Crizotinib | First line | PFS > 15 m |
| 3 | Ou, S.H., et al. [90] | 2014 | (H30:A20) | Crizotinib Alectinib | First line Second line | PD PFS > 12 m |
| 4 | Jang, J. S, et al. [92] | 2016 | (H19:A20) | Not treated with ALK-TKIs | – | Not reported |
| 5 | Couetoux, D. T. M, et al. [94] | 2019 | (H28:A20) | Crizotinib | First line | PR PFS:26.9 m |
| 6 | Tian, P, et al. [89] | 2020 | (H22:A21) | Crizotinib | First line | PR PFS: 7.0 m |
| 7 | Li, M, et al. [24] | 2021 | (H19:A20) | Crizotinib Alectinib | First line Second line | PD PFS > 9 m |
| 8 | Li Y, et al. [91] | 2022 | (H30:A20) | Alectinib | First line | PFS > 19 m |

### 4.4. Other Rare ALK Fusions

In addition to the above rare ALK fusions, there are also some fusions with very low incidence that respond well to ALK-TKIs (Table 4). Cao et al. reported a new form of ALK rearrangement (NCOA1-ALK) in which the patient received PFS for more than 18 months under the treatment of crizotinib [96]. Fang et al. introduced the first lung adenocarcinoma patient carrying myosin phosphatase interacting protein (MPIP) -ALK fusion based on RNA sequencing [97]. Previously, in lung cancer or other tumors, it has been found that MPRIP can interact with neurotrophic tyrosine receptor kinase 1 (NTRK1), platelet-derived

growth factor receptorβ (PDGFRB), and proto-oncogene such as RAF1 [98–100]. An in vitro study found that the MPRIP-ALK fusion gene could promote colony formation, cell growth, and ALK phosphorylation. The corresponding cells were inhibited by crizotinib treatment. Consistent with the study result, the good response of MPRIP-ALK fusion to ALK-TKIs was validated in the clinical presentation of this patient, who was still receiving crizotinib at the time of reporting, with a PFS of at least 11 months. In another multi-sample retrospective study, seven patients were confirmed by NGS sequencing to carry eight rare non-EML4 variants, including CHRNA7-ALK, LOC349160-ALK, TACR1-ALK, KIF5B-ALK, CENPA-ALK, HIP1-ALK, DYSF-ALK and ITGAV-ALK [89]. The seven patients were subsequently treated with crizotinib, with five PRs, two SDs, and a median PFS of 11 months, ranging from 4 to 23 months. Recently, Chen et al. introduced a novel SOS1-ALK fusion [101]. According to related research reports, the son of sevenless homolog 1 (SOS1) gene encodes the SOS1 protein, which is a regulatory protein that is widely expressed in cells. As a key protein in the signaling pathway, SOS1 plays an important role in the regulation of many signal transduction pathways in cells, such as Ras and Rac [102,103]. Abnormal expression or mutation of SOS1 is closely related to the occurrence of clinical diseases. Through clinical observation, this SOS1-ALK fusion (S2:A20) showed a good response to crizotinib, and the patient's PFS exceeded 6 months. A patient with novel LMO7-ALK fusion (L16:A20) reported by Li M et al. rapidly progressed within two months of first-line treatment with alectinib, and then the patient switched to the second-line treatment of ensartinib [71]. Under the preliminary follow-up, PFS has been more than 18 months.

**Table 4.** Published details of other rare ALK fusions positive NSCLC patients treated with ALK-TKIs.

| | Ref | Rare ALK Fusion Types | Merge Mutations | ALK-TKIs | Treatment | Response |
|---|---|---|---|---|---|---|
| 1 | Cao Q, et al. [96] | NCOA1–ALK | CDA$^{K27Q}$, ERCC1$^{N118N}$, DPYD$^{I543V}$, MTHFR$^{A222V}$, GSTP1$^{I105V}$ | Crizotinib | Third line | PFS > 18 m |
| 2 | Fang, W. et al. [97] | MPIP-ALK | | Crizotinib | Second line | PFS > 11 m |
| 3 | Tian, P. et al. [89] | CHRNA7-ALK | | Crizotinib | First line | PFS: 18 m |
| 4 | Tian, P. et al. [89] | LOC349160-ALK | | Crizotinib | First line | PFS: 7 m |
| 5 | Tian, P. et al. [89] | TACR1-ALK | | Crizotinib | First line | PFS: 15 m |
| 6 | Tian, P. et al. [89] | CENPA-ALK | | Crizotinib | First line | PFS: 4 m |
| 7 | Tian, P. et al. [89] | DYSF-ALK ITGAV-ALK | ALK p.Q1146P; MET p.M636V | Crizotinib | First line | PFS: 23 m |
| 8 | Chen, H.F. et al. [101] | SOS1-ALK | | Crizotinib | First line | PFS > 6 m |
| 9 | Li, M. et al. [71] | LMO7-ALK | NRG1 c.602A > T;TP53 | Alectinib Ensartinib | First line Second line | PD PFS > 18 m |

The patients who harbor double ALK fusion variants are extremely rare. Few investigations have focused on the concomitance of double ALK rearrangements because of the low incidence (Table 5). According to our literature search results, only eleven cases have been previously reported, including CCNY-ALK and ATIC-ALK [104], NLRC4-ALK and EML4-ALK [105], PRKCB-ALK and EML4-ALK [106], EML4-ALK and BCL11A-ALK [107], EML6-ALK and FBXO11-ALK [108], DYSF-ALK and ITGAV-ALK [109], ALK-SSH2 and EML4-ALK [110], ARID2-ALK and EML4-ALK [110], EML4-ALK and CDK15-ALK [111], PDK1-ALK and STRN-ALK [68], as well as ALK-GCA and EML4-ALK [112], etc. Previous reports confirmed that patients with double ALK fusion may respond to ALK-TKIs. However, the responses are heterogeneous for patients with different ALK fusions. The effectiveness of ALK-TKI treatment might be affected by the two kinds of ALK mutations exist simultaneously in one patient. In other words, the disappearance of one ALK fusion in patients with double ALK fusion may be the reason affecting the therapeutic effect of ALK-TKIs. Additionally, one study speculated that coexistence of double ALK fusion may be related to the occurrence of serious adverse events or drug resistance.

**Table 5.** Published details of other double ALK fusion positive NSCLC patients treated with ALK-TKIs.

| | Ref | Year | Double Fusion | ALK-TKIs | Treatment | Response |
|---|---|---|---|---|---|---|
| 1 | Wu, X. et al. [104] | 2020 | CCNY-ALK ATIC-ALK | Crizotinib | First line | PR > 6 m |
| 2 | Wu, X. et al. [105] | 2020 | NLRC4-ALK; EML4-ALK | Crizotinib | First line | PFS > 10 m |
| 3 | Luo, J. et al. [106] | 2019 | PRKCB-ALK; EML4-ALK | Crizotinib Ceritinib | First line Second line | PFS: 6 m PFS > 2 m |
| 4 | Qin, B. et al. [107] | 2019 | BCL11A-ALK; EML4-ALK | Crizotinib | First line | PFS: 13 m |
| 5 | Lin, H. et al. [108] | 2018 | FBXO11-ALK; EML6-ALK | Crizotinib | Second line | PFS > 11 m |
| 6 | Yin, J. et al. [109] | 2018 | DYSF-ALK; ITGAV-ALK | Crizotinib | Second line | PFS > 3 m |
| 7 | Tao, H. et al. [110] | 2022 | ALK-SSH2; EML4-ALK | Crizotinib | First line | PFS: 9 m |
| 8 | Tao, H. et al. [110] | 2022 | ARID2-ALK; EML4-ALK | Crizotinib | First line | PFS: 12 m |
| 9 | Guo, J. et al. [111] | 2020 | CDK15-ALK; EML4-ALK | Crizotinib | Second line | PFS: 23 m |
| 10 | Zeng, H. et al. [68] | 2021 | PDK1-ALK; STRN-ALK | Alectinib | First line | PFS > 11 m |
| 11 | Zhai, X. et al. [112] | 2022 | ALK-GCA; EML4-ALK | Alectinib | First line | PFS > 20 m |

However, not all rare fusion types are sensitive to ALK-TKIs. Previous studies have shown that fusion partners need to provide dimerization domains to facilitate the automatic activation of kinases. Similar to other tyrosine kinases, ALK must dimerize to automatically activate and signal downstream [1]. Many fusion partners contain coiled coil domains or other known dimerization domains, but not all fusion partners have obvious dimerization motifs. PTPN3-ALK is predicted to be unresponsive to crizotinib treatment because it lacks the ALK kinase domain [113]. In addition, studies have also reported that CMTR1-ALK does not respond to crizotinib. It is hypothesized that this particular type of ALK fusion is a null fusion and thus unable to translate the kinesin that causes tumorigenesis [114].

## 5. Conclusions

Since the discovery of the EML4-ALK fusion in NSCLC, a variety of ALK-TKIs have been developed to treat ALK-positive NSCLC. The rapid development of targeted therapy has resulted in significant improvements in PFS and OS in patients with metastatic ALK-positive NSCLC. However, the heterogeneity of clinical response exists not only among different ALK fusion subtypes, but also among different variants. Among the rare fusions of ALK, we found that even in homozygous fusions, different variants responded differently to ALK-TKIs.

There are two possible explanations for the heterogeneity of responses to ALK-TKIs: one is that different fusion partners lead to differences in protein stability and expression levels, and the other is other genetic alterations that accompany ALK rearrangements leading to different responses to ALK-TKIs. In this article, we summarize that different 5′ partners affect the biological properties of ALK fusion proteins, including kinase activity, protein stability, transformation potential, and most importantly, the response to ALK-TKIs. Patients with rare ALK fusion can combine the type of ALK fusion and the resistance or sensitivity of existing mutations to different ALK-TKIs, and flexibly combine the first-, second-, and third-generation ALK inhibitors for sequential treatment to improve the prognosis.

FISH, as the gold standard for detecting ALK-positive, cannot identify specific fusion forms. Since different variants may have different responses to ALK-TKIs, it is critical to identify specific variants in different individuals to enable precision drug therapy in the future. To some extent, NGS may be a better complementary detection method. With the application and popularization of NGS technology, more and more rare ALK fusions have been discovered one after another, helping ALK-positive NSCLC patients receive more precise targeted therapy. At the same time, NGS can also detect the ctDNA and cfDNA in the blood to predict the therapeutic effect and drug resistance mechanism of the drug, thereby improving the prognosis of patients with ALK-positive NSCLC.

However, due to the limited number of rare fusion cases, it is difficult to compare the reasons for the differential responses of different rare fusions to ALK-TKIs and their resistance mechanisms. In the future, genomic, transcriptomic, and proteomic analyses are needed to investigate overall therapeutic strategies for rare fusions in ALK. At the same time, clinicians are also encouraged to report these novel fusions and provide information on fusion breakpoints and responses to ALK-TKIs to better understand the application of ALK-TKIs in rare ALK rearrangements.

**Author Contributions:** Conceptualization, Y.X. and K.L.; methodology, S.Z.; software, X.F.; validation, Y.X. and S.Z.; formal analysis, Y.J.; resources, K.L.; data curation, J.L. and T.F.; writing—original draft preparation, Y.X.; writing—review and editing, S.Z.; visualization, J.L.; supervision, T.F.; project administration, K.L.; funding acquisition, K.L. All authors have read and agreed to the published version of the manuscript.

**Funding:** This study is supported by the National Natural Science Foundation of China (82172708).

**Conflicts of Interest:** The authors declare that the research was conducted in the absence of any commercial or financial relationships that could be construed as a potential conflicts of interest.

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
