# Peer review of "Therapeutic Advances of Rare ALK Fusions in Non-Small Cell Lung Cancer"

_curroncol, doi:10.3390/curroncol29100618_

Round 1
Reviewer 1 Report
Recommendation: Major
In this review, the authors introduce the biological characteristics of ALK fusion kinase and common detection methods of ALK fusion, focus on summarizing the differential responses of several rare ALK fusions to ALK-TKIs, and propose corresponding treatment strategies, so as to better guide the application of ALK-TKIs in rare ALK fusion population. This review should be greatly improved before considering its publication. Some specific comments should be pointed out.
1. The authors should give the biological characteristics of ALK fusion kinase and the pictures of common ALK fusion detection methods at an appropriate position in the paper, so that readers can more clearly understand the principle of rare ALK fusion therapy for non-small cell lung cancer.
2. In the paper, the authors did not provide information about the public details of patients with KIF5B-ALK-positive non-small cell lung cancer receiving ALK TKIs treatment.
3. In the paper, the authors did not write about the reasons for different reactions of different rare fusions to ALK TKI and their resistance mechanisms, but only provided a large number of patients with ALK fusion-positive non-small cell lung cancer treated with ALK TKIs in detail, which would be more perfect if added slightly.
4. The authors need to focus more on the research development in recent 5 years. In the manuscript, about one-third of references are over 5 years. I suggest the authors adjust these references.
Author Response
All responses are in the attachment.

Reviewer 2 Report
Yan and co-author summarize the therapeutic advances of rare ALK fusions in non-small cell lung cancer, which is important and helpful to understand the target therapy. Recommend publish after minor revision
1, There are many reviews about ALK fusion and testing methods. The author should summarize and highlight in this manuscript that what’s the context that other review did not summarize in it in the introduction.
2, lowcase of T of “The” in the line 11,
3, add blank after dot in line 55, 58, 75, 82,…, The author should check the entire manuscript, and add blank in many other places.
4, The author should uniform the font size, in line 99-107.
5, Uppercase the Fish to FISH in the line 124.
6, The authors should separate the words in the subtitle of 3.3 “3.3Reversetranscriptionpolymerasechainreaction.” to “3.3 Reverse Transcription polymerase chain reaction”
7, The author should cite the Table 2 in the main text where it should be.
8, In the other rare ALK fusion section, the author should cite the recent references about the novel rare ALK fusions. CDK15-ALK, PDK1-ALK and some of other double fusions.
Author Response
Response 1:Thank you for your suggestion. The summary of ALK testing methods has been added to the introduction. We have roughly summarized the characteristics of each method, such as: As far as ALK gene detection is concerned, Fluorescence in situ hybridization (FISH) is the preferred gold standard method; Immuno-histochemistry (IHC) can be used for screening because it is simple and cheap, and the FDA-approved antibody (Ventana D5F3) can be used to detect ALK fusion independently without FISH validation. Reverse transcription polymerase chain reaction (RT-PCR) can also be used, but it cannot detect some rare fusions. Next generation sequencing (NGS) can not only detect ALK fusion, but also determine the type of fusion and other accompanying driver genes, which meets the current clinical needs. We put the specific contents of detection methods in 3. Detection methods for ALK rearrangements, thank you!
Response 2:I apologize for our mistakes, we have revised the question you mentioned and double checked the full text.
Response 3:I apologize for our mistakes, we have revised the question you mentioned and double checked the full text.
Response 4:I apologize for our mistakes. We have corrected the formatting problem you mentioned and checked the full text again.
Response 5:I apologize for our mistakes. We have corrected the formatting problem you mentioned and checked the full text again.
Response 6:I apologize for our mistakes because of not separating the words in the subtitle of 3.3 "3.3 Reverse Transcription Polymerase Chain Reaction". We have corrected this low-level error and checked the full text again.
Response 7:We have checked all the figures and tables in this review and correctly cited them in the appropriate places in this review. Thank you for your correction.
Response 8:Thank you very much for your suggestion, which makes our review more comprehensive. Some rare ALK double fusion such as CDK15-ALK, PDK1-ALK and other double fusion have been added to the text. Specific contents: the patients who harbor double ALK fusion variants are extremely rare. Few investigations have focused on the concomitance of double ALK rearrangements because of the low incidence (Table 6). According to our literature search results, only eleven cases have been previously reported, including CCNY-ALK and ATIC-ALK 、NLRC4-ALK and EML4-ALK 、PRKCB‐ALK and EML4‐ALK 、EML4‐ALK and BCL11A‐ALK、EML6-ALK and FBXO11-ALK、DYSF‐ALK and ITGAV‐ALK、ALK-SSH2 and EML4-ALK、ARID2-ALK and EML4-ALK、EML4-ALK and CDK15-ALK 、PDK1-ALK and STRN-ALK、as well as ALK-GCA and EML4-ALK,etc. Previous reports confirmed that patients with double ALK fusion may respond to ALK-TKIs. However, the responses are heterogeneous for patient with different ALK fusions. Effectiveness of ALK‐TKIs treatment might be affected by the two kinds of ALK mutations exist simultaneously in one patient. In other words, the disappearance of one ALK fusion in patients with double-ALK fusion may be the reason affecting the therapeutic effect of ALK-TKIs. Additionally, one study speculated that coexistence of double ALK fusion may be related to the occurrence of serious adverse events or drug resistance.
Round 2
Reviewer 1 Report
Accept in present form